# Increased Plasma Levels of Adenylate Cyclase 8 and cAMP Are Associated with Obesity and Type 2 Diabetes: Results from a Cross-Sectional Study

**DOI:** 10.3390/biology9090244

**Published:** 2020-08-24

**Authors:** Samy M. Abdel-Halim, Ashraf Al Madhoun, Rasheeba Nizam, Motasem Melhem, Preethi Cherian, Irina Al-Khairi, Dania Haddad, Mohamed Abu-Farha, Jehad Abubaker, Milad S. Bitar, Fahd Al-Mulla

**Affiliations:** 1Department of Oncology, Karolinska University Hospital, 17177 Stockholm, Sweden; samy.abdel-halim@sll.se; 2Department of Genetics and Bioinformatics, Dasman Diabetes Institute, Dasman 15462, Kuwait; rasheeba.iqbal@dasmaninstitute.org (R.N.); motasem.melhem@dasmaninstitute.org (M.M.); dania.haddad@dasmaninstitute.org (D.H.); 3Department of Biochemistry and Molecular Biology, Dasman Diabetes Institute, Dasman 15462, Kuwait; preethi.cherian@dasmaninstitute.org (P.C.); irina.alkhairi@dasmaninstitute.org (I.A.-K.); mohamed.abufarha@dasmaninstitute.org (M.A.-F.); jehad.abubakr@dasmaninstitute.org (J.A.); 4Department of Pharmacology and Toxicology, Faculty of Medicine, Kuwait University, Jabriya 046302, Kuwait; milad.bitar@gmail.com

**Keywords:** adenylate cyclase 8, ADCY8, ADCY3, ADCY1, Type 2 diabetes, obesity, cAMP

## Abstract

Adenylate cyclases (ADCYs) catalyze the conversion of ATP to cAMP, an important co-factor in energy homeostasis. Giving ADCYs role in obesity, diabetes and inflammation, we questioned whether calcium-stimulated ADCY isoforms may be variably detectable in human plasma. We report the results of a cross-sectional study assessing circulating levels of functional ADCY1, −3 and −8 in patients with T2D vs. non-diabetic (ND) controls in association with obesity. ADCY1 levels exhibited no significant change between ND and T2D groups. ADCY3 levels were lower in obese individuals, albeit not statistically significantly. In contrast, ADCY8 plasma levels were significantly higher in obese and T2D patients compared to controls (*p* = 0.001) and patients with T2D only (*p* = 0.039). ADCY8 levels correlated positively with body mass index and Hb1Ac levels. Parallel to the increased ADCY8 levels, significantly higher cAMP levels were observed in patients with T2D compared with ND controls, and further elevated in obese individuals, irrespective of T2D status. Additionally, cAMP levels positively correlated with fasting plasma glucose levels. In conclusion, the current cross-sectional study demonstrated elevated levels of circulating plasma ADCY8 and cAMP in obesity and T2D.

## 1. Introduction

The prevalence of obesity continues to surge globally and a major cause of morbidity, mortality, and low quality of life [1]. Obesity is a systemic disease caused by abnormal accumulation of body fat, which triggers chronic low-grade inflammation resulting in diverse complications including diabetes, angiopathy, and cardiovascular disease [2,3,4]. Type 2 diabetes (T2D) is a heterogeneous condition characterized by impaired insulin secretion in response to glucose stimulation and development of insulin resistance [5,6,7]. The increased incidence of obesity and T2D and associated complications is of global public health concern and exerts a heavy burden on health system providers, especially in the developing world [8,9]. Hence, identifying simple blood or plasma biomarkers that enable the prediction and monitoring of diabetes is of paramount importance.

Adenylate cyclases (ADCYs) are membrane-bound enzymes, which catalyze the conversion of adenosine triphosphate (ATP) to cyclic adenosine monophosphate (cAMP) [10], which is critical to maintaining energy homeostasis. In humans, ten ADCY isoforms have been identified and classified into various families according to their mode of stimulation [11,12,13,14]. Of these, three calcium-stimulated isoforms (ADCY1, −3 and −8) were grouped as a family of rapid responders [11,14,15]. In animal models of T2D, studies have identified ADCYs as causal drivers of obesity and diabetes. The expression of ADCY3 in pancreatic islets and its correlation with insulin secretion in T2D were first discovered in one of our previous studies, which demonstrated overexpression of ADCY3 in islets from a T2D animal model caused by the presence of two functional point mutations in the ADCY3 gene promoter [16]. In addition, ADCY3-knockout mice were found to be more vulnerable to obesity induced by a high-fat diet [17]; meanwhile, the gain-of-function ADCY3^M279I^ mutant protects animals from diet-induced metabolic imbalance [18]. In humans, the ADCY3 gene was cloned, sequenced and was demonstrated to be widely expressed in several tissue types, including brain, skeletal muscle, heart, kidney, liver, pancreatic islets, and placenta, all known to be significantly modulated by serum glucose levels in the diabetic state [19]. Interestingly, early onset of monogenic obesity is associated with homozygous mutations within the human ADCY3 gene [20], highlighting the role of ADCY3 in energy homeostasis [21]. Studies on expression patterns of ADCY isoforms identified ADCY8 to be distinctly overexpressed in diabetic islets [22,23] and adrenal glands [24]. Furthermore, G-protein coupled to calcium-mediated ADCY signaling, namely G α-s and G α-olf, was also shown to be overexpressed in an animal model of spontaneous T2D [23]. Nevertheless, the role of ADCY8 in functional pancreatic islets is debatable. While some studies have demonstrated the importance of ADCY8 in restoring glucose-induced insulin secretion and in mediating the effects of glucagon-like peptide 1 (GLP-1) [25,26], others have failed to detect ADCY8 transcripts in human islets [27,28].

Taken together, these findings suggest a potential role for circulating plasma levels of ADCY in whole body energy and glycemic homeostasis. Here, we quantified plasma levels of ADCY proteins and cAMP in patients with diabetes and obesity, aiming to investigate their potential role in the prediction and monitoring of obesity and T2D.

## 2. Results

### 2.1. Demographic Data of Study Population

The characteristics of the 188 individuals included in this study are listed in Table 1. Patients with T2D were older, and had a significantly higher weight, body mass index (BMI), and waist-to-hip ratio than controls. The female-to-male ratio was higher among individuals without T2D than among those with T2D. Patients with T2D had higher levels of fasting plasma glucose (FPG), glycated hemoglobin (HbA1c), insulin and triglyceride (TGL) compared with controls (*p* < 0.05). The study population was further subdivided according to obesity status (Table 2). Among patients with T2D, obese individuals had significantly higher FPG and HbA1c levels compared with non-obese individuals (*p* = 0.018 and *p* < 0.001, respectively).

### 2.2. Plasma Levels of ADCY Proteins, Obesity and Diabetes

Next, we investigated the relationship between the different ADCYs and clinical characteristics of the participants. As observed in Figure 1A, relative to ND controls, patients with T2D had statistically significant elevated plasma levels of ADCY8 (mean ± SD: 12.645 ± 1.69 vs. 12.056 ± 1.64 ng/mL, *p* = 0.017). Likewise, two-way ANOVA with the post-hoc Tukey test revealed a significant increase in ACDY8 plasma levels in patients with T2D compared with ND controls (*p =* 0.04). In contrast, ADCY1 and ADCY3 levels in patients with T2D did not differ significantly from those of ND controls, even when two-way ANOVA was applied. When ADCY levels were investigated in relation to obesity status, no significant differences were observed between obese and non-obese participants (Figure 1B).

Interestingly, when the data were stratified based on obesity status (Figure 2), a statistically significant increase in ADCY8 plasma levels was detected in obese patients with T2D (12.941 ± 1.548) relative to non-obese patients with T2D (12.142 ± 1.820) (*p* = 0.039). In addition, two-way ANOVA with the post-hoc Tukey analysis showed that ADCY8 plasma levels did not differ in relation to obesity per se (*p* > 0.05); however, the levels were significantly increased in obese-diabetic patients (*p* = 0.02). Taken together, these data may indicate a synergistic effect of obesity and diabetes factors that enhances plasma ADCY8.

In contrast, no significant differences in the plasma levels of ADCY1 or ADCY3 were observed between obese and non-obese patients, irrespective of T2D diagnosis (Figure 2). Notably, the low variability in ADCY3 plasma levels observed in patients with T2D might be attributed to age. The patients with T2D enrolled in our study were significantly older than ND controls (*p* < 0.001, Table 1); concomitantly, ADCY3 is positively correlated with the age factor in ND controls, but not in patients with T2D (Appendix A).

In the ND controls, the ratio of females to males was 2.5:1 (Table 1); however, no gender influence was associated with the plasma levels of the studied ADCYs (Appendix A). Similarly, when the study participants were stratified by participant age (<46 vs. ≥46 years old), the distribution of ADCY data was notably modified as observed in Appendix A. The ADCY1 levels in younger individuals (<46 years) were marginally higher than those of the older individuals (*p* = 0.02).

### 2.3. Plasma Levels of ADCY Proteins and Biomedical Parameters

We further investigated the correlations between ADCY levels and biomedical parameters. To this end, plasma ADCY8 levels exhibited a significant positive correlation with plasma Hb1Ac (r = 0.188, *p* = 0.012) and BMI (r = 0.151, *p* = 0.040, Figure 3). While the correlation between ADCY8 and Hb1Ac was observed to be consistent after adjusting for age and sex, the correlation between ADCY8 and BMI was compromised (Appendix A). Meanwhile, ADCY1 levels positively correlated with insulin levels (r = 0.242, *p* = 0.008) and TGL (r = 0.201, *p* = 0.008; Figure 3), and these findings were consistent after adjusting for age and sex. In contrast, ADCY3 levels failed show significant correlation with any of the tested biomedical parameters; however, after adjusting to the confounders, an inverse correlation was observed between ACDY3 and BMI (r = −0.183, *p* = 0.04, Appendix A).

Then, the cohort was subdivided based on being diabetic or obesity status, ADCY8 plasma level showed a significant positive correlation with BMI (r = 0.294, *p* = 0.005, Appendix A) in T2D patients, but not in ND; whereas, plasma ADCY8 and HbA1C were significantly correlated obese participants only (r = 0.306, *p* = 0.002, Appendix A). On the other hand, ADCY1 showed a significant positive correlation with insulin and TGL in both T2D patients only (r = 0.326, *p <* 0.007) and obese individuals (insulin (r = 0.357, *p* = 0.006) and TGL (r = 0.247, *p* = 0.018), Appendix A). Under this data setup, ADCY3 showed significant correlation with FPG in ND controls only (r = 0.238, *p* = 0.032). The previously observed inverse correlation between ADCY3 and BMI was further observed in obese individuals (r = −0.3204, *p* = 0.005, Appendix A).

### 2.4. cAMP Plasma Levels

ADCY protein plasma levels may not reflect their functionality. To determine the functional implication of the higher ADCY8 plasma levels observed in this study, we quantified the circulating plasma levels of its end-product, namely the secondary messenger cAMP, in our study population (Figure 4). Patients with T2D exhibited significantly higher levels of plasma cAMP compared with ND controls (mean ± SD: 28.287 ± 12.290 vs. 16.707 ± 9.251 pmol/mL, *p* < 0.001; Figure 4A). Obese individuals had higher levels of cAMP, both among ND controls (mean ± SD: 26.436 ± 8.474, *p* < 0.001) and among T2D patients (mean ± SD: 31.062 ± 13.608, *p* < 0.001; Figure 4B). Two-way ANOVA further indicated that obesity alone tended to have a more profound influence on plasma cAMP levels (*p* < 0.005), suggesting a protective role of hyperinsulinemia in these individuals.

In the entire cohort, we studied prospective correlations between biomedical parameters and the circulating cAMP levels. Of all the studied parameters, cAMP showed a positive correlation with FPG (r = 0.266, *p* = 0.039) and BMI (r = 0.297, *p* = 0.017) (Figure 4C,D). These data may suggest a functional role for the increases in ADCY8 levels in patients with T2D, glucose metabolism, and obesity and warrant urgent and larger cross-sectional studies to be planned in the future.

## 3. Discussion

To the best of our knowledge, this is the first study quantifying circulating levels of rapid-responder calcium-mediated ADCY isoforms in human plasma. Previously, we demonstrated ADCY overexpression in animal model of spontaneous T2D [16,22,23,29]. In those studies, we observed a distinct overexpression of ADCY8 in the islets of diabetic animals but not in control animals [22]. These findings, together with the fact that ADCYs are widely expressed in several tissue types, suggested the potential availability of ADCYs in the blood circulation. The results of the current study corroborated this hypothesis and our earlier observations in T2D animals by finding similarly significantly enhanced levels of both ADCY8 and its functional secondary messenger, cAMP, in plasma from patients with T2D compared with ND controls. The levels of this signaling cascade were demonstrated to be even higher in diabetic patients with obesity.

In the present study cohort, ADCY1 plasma levels did not differ significantly, irrespective of T2D diagnosis or obesity status. However, correlation analyses with clinical parameters revealed a positive correlation between plasma ADCY1 levels, TGL and insulin, suggesting a potential association with dyslipidemia and a prediabetic state, respectively. In obese participants, plasma ADCY3 levels and BMI were inversely correlated, which is in accordance with two recent population studies, identifying loss-of-function mutations within the ADCY3 gene are associated with severe obesity and T2D [20,30]. In addition, ADCY3 gene polymorphisms were shown to be associated with obesity in patients with T2D; however, in this study, the impact of the gene variants on ADCY3 plasma or cellular levels were not explored [31]. Subsequently, Wang et al. demonstrated that ADCY3-deficient mice were prone to obesity [17]. On the other hand, hepatic ADCY3 upregulation by liraglutide was also shown to reduce body weight and improve insulin resistance in a mouse model [32]. Taken together, our data is in accordance with these reports, indicating a role for ADCY3 in obesity and T2D. A larger study is now being performed in our laboratory to investigate this in more detail.

ADCY8 plasma levels were higher in patients with T2D, and levels were further increased in obese T2D patients. Furthermore, the positive correlation between ADCY8, BMI and HbA1c levels suggests that both obesity and poor glycemic control may be the contributing factors in this regard. Several reports suggested a role of ADCY8 in T2D. Previously, ADCY8 was determined to be solely overexpressed in the pancreatic β- and α-cells in diabetic animal models of T2D but not in islets from control rats [22], whereas ADCY1 and −3 are expressed in the islets of both control and diabetic rodents [23]. A recent genome-wide analysis study identified an association between ADCY8 gene variants, obesity, and abnormal adipose tissue depots [33]. The modulation effect of prevailing glucose levels on ADCY activity was previously demonstrated [29]. Glucotoxicity was reported to downregulate ADCY8 in pancreatic β-cell lines [25,26]. Compared with other ADCYs, ADCY8 is of particular interest, as it is a target for the synergy between the GLP-1 receptor and glucose signaling [34], and an important regulator of glucose tolerance and hypothalamic adaptation to high-fat diet via regulation of islet insulin secretion [26]. Notably, the positive correlation between ADCY8 and BMI in obese T2D patients, but not in obese ND, may imply that obesity per se is not sufficient to induce an increase in ADCY8 plasma levels; the presence of diabetic state and poor glycemic control is necessary for enhancing ADCY-cAMP levels in obese individuals.

The source of circulating ADCYs is not known. We hypothesize that since ADCYs are membranous proteins, it is possible that circulating ADCYs may be released into the circulation as a part of the exosome system. In support of this hypothesis, proteomic profiling of exosomes purified from human plasma identified ADCY10 as a component of the exosome fraction (supplementary data in [35]). In addition, ADCY1 and ADCY4 were detected in human urinary extracellular vesicles (supplementary data in [36]). The role of ADCYs in the plasma or exosome system has not been well studied and should be addressed in future studies.

Relative to ND controls, we observed higher levels of plasma cAMP in T2D patients and obese individuals. In addition, cAMP levels exhibited a positive correlation with FPG, HbA1c, and BMI; hence, these data suggest an association between plasma cAMP and glycemic homeostasis and obesity. Studies have shown that adipose tissue development and function is dependent on cAMP signaling, which regulates genes related to adipogenesis, lipolysis, and thermogenicity [37,38,39]. In addition, elevated cAMP levels were previously demonstrated to be associated with the development of complications in T2D [40]. For example, in cardiac arrhythmia, cAMP initiates arrhythmogenic signals on its own [27] or promotes arrhythmia indirectly through the phosphorylation of Ca^2+^/calmodulin signaling cascades [41]. Taken together, we propose that establishing the plasma levels of ADCY8 or cAMP may have direct and relevant implications on patient care and development of obesity and T2D complications. This hypothesis remains to be validated in large prospective studies.

## 4. Conclusions

In conclusion, plasma levels of ADCY8 were found to be elevated in patients with obesity, and even more so in the obese-T2D group. Further studies are currently underway in our laboratory to understand the consequences of high ADCY 8 plasma levels in obese-T2D individuals in terms of cardiac arrhythmias and other complications. Larger population-wide studies are warranted to further clarify the association between circulating plasma levels of ADCY proteins in T2D/obesity and their clinical significance.

## 5. Materials and Methods

### 5.1. Study Design

This was a cross-sectional cohort study carried out to quantify the levels of circulating plasma ADCY1, −3 and −8 and their activity as a function of cAMP plasma levels in obese individuals diagnosed with T2D vs. controls.

### 5.2. Study Population

The study cohort comprised 188 individuals, including 92 patients with T2D and 96 individuals with no previous T2D diagnosis, as assessed by an FPG < 5.5 mmol/L in accordance with the American Diabetes Association, ADA, criteria. Obesity was defined as having a BMI ≥ 30 kg/m^2^; where, BMI was determined by specialized nurse at Dasman Diabetes Institute (DDI), Kuwait, using the standard formula for BMI (BMI = body weight [kg]/height^2^ [m^2^]). Written informed consent was obtained from all study participants following the ethical guidelines of the Declaration of Helsinki and the study was approved by the Ethical Review Board of DDI. Diabetic patients were recruited from DDI outpatient clinic. The healthy controls were among individuals enrolled at the gymnasium facility or other types of exercise studies conducted by researchers at DDI. The study design described inn our previous publications [42,43]. Participants with chronic T2D and complications were excluded from the study. Additionally, individuals with prior major illness or taking any medication and/or supplement known to influence the body composition, bone mass, insulin action or insulin secretion and/or pancreatic β-cell function were excluded from the study. Furthermore, individuals who were morbidly obese (BMI ≥ 40 kg/m^2^) and those diagnosed with type 1 diabetes (T1D) were also excluded.

### 5.3. Blood Collection, Anthropometric, and Biochemical Measurements

Blood samples were collected from the study subjects, and plasma was prepared using vacutainer EDTA tubes, aliquoted and subsequently stored at −80 °C until use as described previously [44]. FPG, triglyceride, total cholesterol, low-density lipoprotein (LDL), and high-density lipoprotein (HDL) were measured on the Siemens Dimension RXL chemistry analyzer (Diamond Diagnostics, Holliston, MA, USA) [45]. Glycated hemoglobin (HbA1c) was measured using the Variant^TM^ device (Bio-Rad, Hercules, CA, USA). Insulin resistance was assessed using the homeostatic model assessment of insulin resistance (HOMA-IR) formula(FPG [mmol/L] × fasting insulin [mU/L]/22.5), as previously described [46].

### 5.4. Quantitative Analysis of Plasma Levels of Adenylate Cyclase Proteins

ADCY1, ADCY3 and ADCY8 plasma levels were measured using an ELISA kit from MyBiosource, San Diego, CA, USA. The assay was used according to the instructions of the manufacturer. Plasma samples were thawed on ice and centrifuged at 10,000× *g* for 5 min at 4 °C to remove debris. Serial dilutions of the plasma sample were performed to determine the optimal dilution factor for each kit. Based on the kit’s specifications, spiking analysis resulted in 92–101% recovery for each of the ADCY proteins, evincing high specificity and no interaction between other analogs.

### 5.5. Quantitative Analysis of cAMP Plasma Levels

Specific ELISA kits (Cayman Chemical Company, Ann Arbor, Michigan, USA) were used to quantify plasma cAMP levels following the manufacturer’s instructions. Plasma samples and standards were acetylated prior to analysis, and total cAMP was detected using the manufacturer’s recommended standard curve with a range of 0.078–10 pmol/mL. The average of three optical density (OD) readings at 420 nm was recorded, and the data were plotted as logit (B/B0) vs. log concentration, adopting a linear regression fit curve. Results were expressed as pmol/mL.

### 5.6. Statistical Analysis

The Shapiro Wilks test for normality was used to evaluate the distribution of variables (Appendix A). Comparisons between study groups were made by two-tailed, unequal variance *t*-tests and Pearson’s correlation coefficient test for normally distributed data. Non-parametric tests (Mann–Whitney test and Spearman’s correlation coefficient) were used for data reflecting variables. A two-way ANOVA with the post-hoc Tukey test was performed to evaluate the effect of diabetes and obesity on the expression of ADCY proteins. All data are reported as mean ± standard deviation (SD). All statistical calculations were two-sided, and differences were considered significant when the probability value (*p*) was < 0.05. All analyses were performed using IBM SPSS Statistics for Windows, Version 25.0. Armonk, NY: IBM Corp.

## Figures and Tables

**Figure 1 biology-09-00244-f001:**
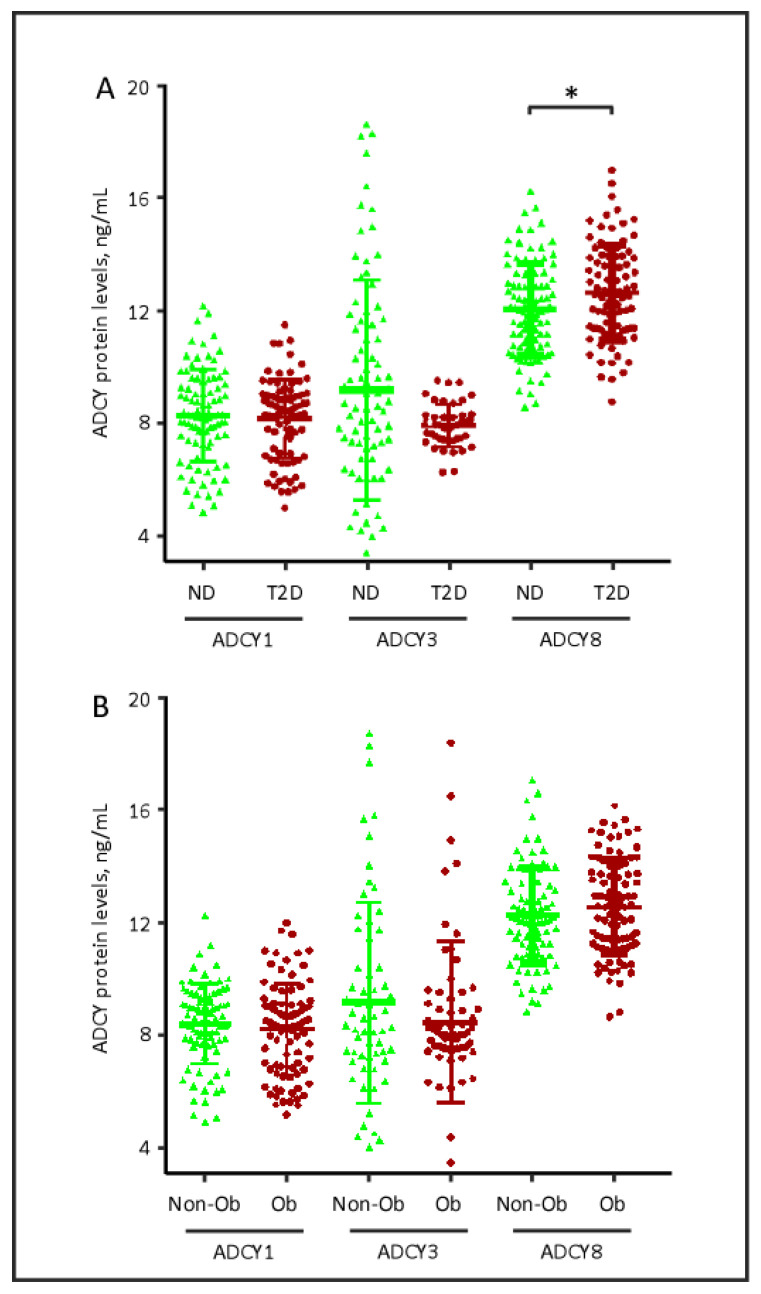
Quantitative analysis of plasma ADCY1, ADCY3 and ADCY8 in (**A**) subjects with and without T2D and in (**B**) subjects with and without obesity. Data are presented as mean ± SD. ADCY, adenylate cyclase; ND, non-diabetic; T2D, type 2 diabetes; Non-Ob, non-obese; Ob, Obese. * *p* = 0.017). *p*-Value was calculated using unequal variance *t*-test.

**Figure 2 biology-09-00244-f002:**
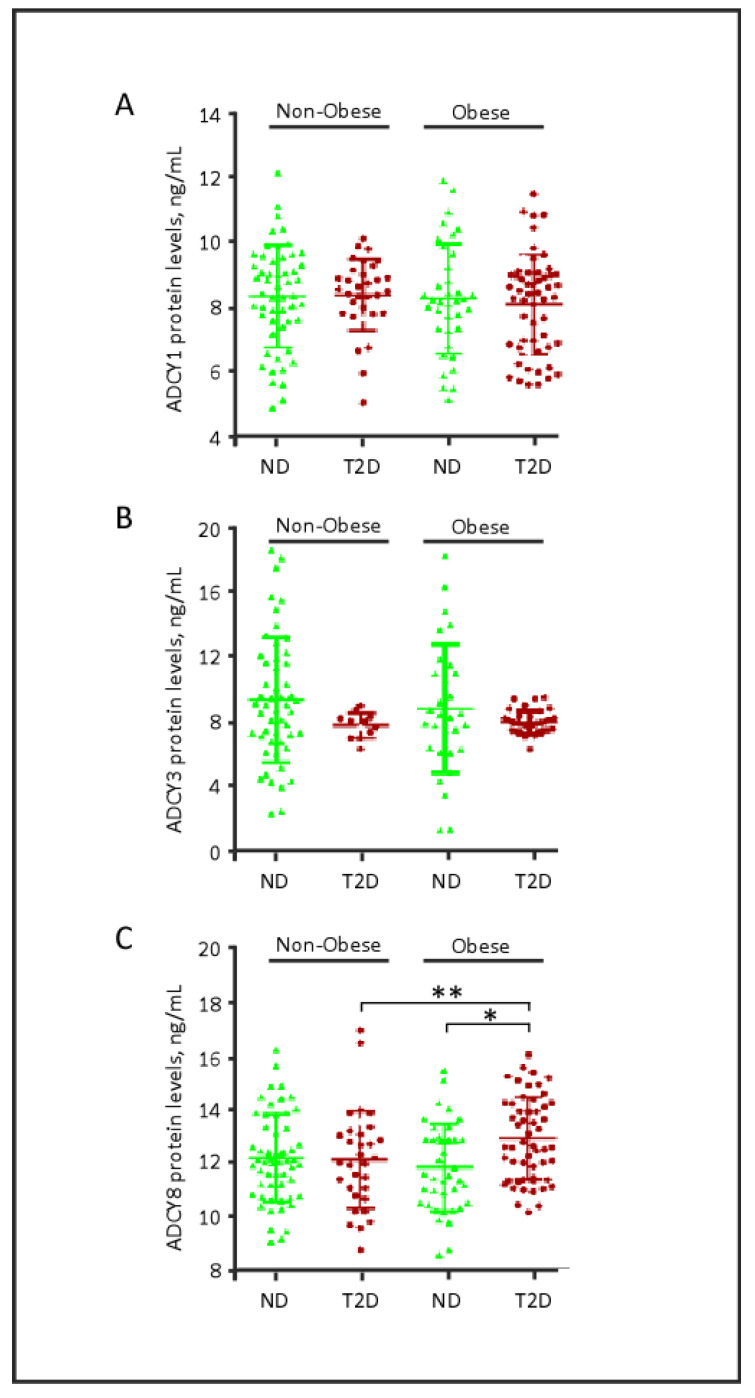
Quantitative analysis of ADCY1, ADCY3 and ADCY8 based on diabetes and obesity status. * *p* = 0.001, ** *p* = 0.039. The statistical tests are Mann Whitney U test for ADCY3 (**B**) and unequal variance *t*-test for ADCY1 (**A**) and ADCY8 (**C**). ND, non-diabetic; T2D, type 2 diabetes.

**Figure 3 biology-09-00244-f003:**
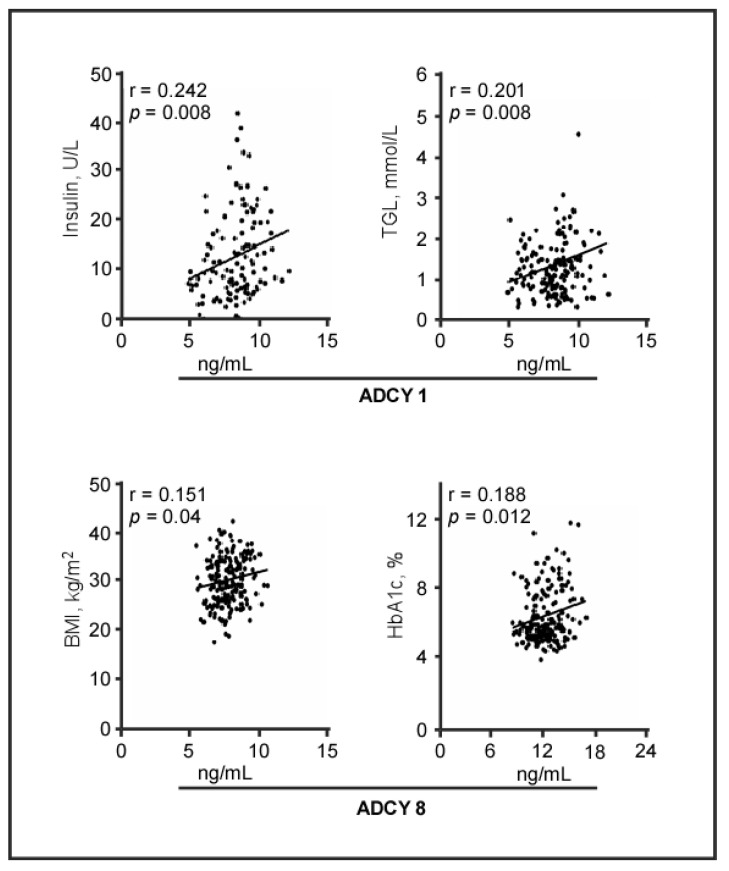
Correlation between plasma levels of ADCY isoforms and biochemical variables. Correlation coefficients (r) were calculated using Pearson’s correlation. ADCY, adenylate cyclase; BMI, body mass index; HbA1c, glycated hemoglobin; r, correlation coefficient; TGL, triglycerides.

**Figure 4 biology-09-00244-f004:**
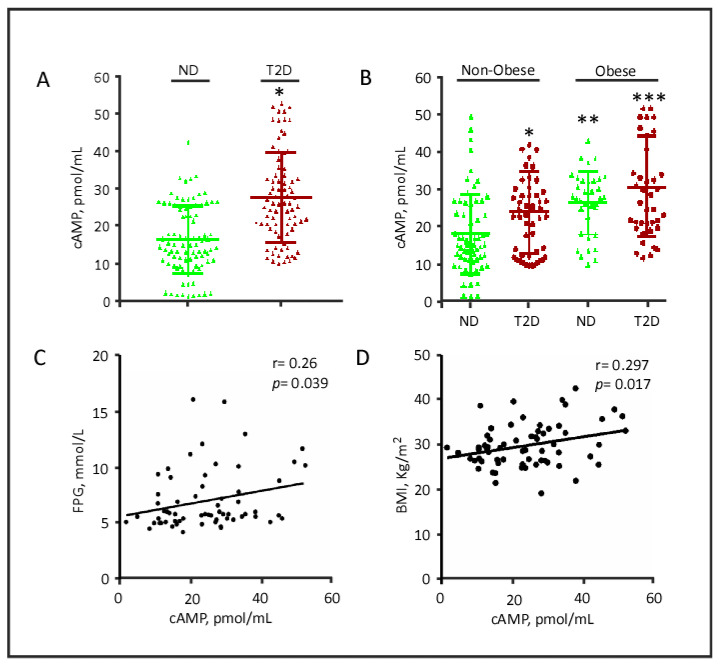
Plasma levels of cAMP in the study population. (**A**) Plasma levels of acetylated cAMP in subjects with and without T2D. (**B**) Plasma levels of acetylated cAMP in obese and non-obese subjects, with and without T2D. Data are presented as mean ± SD. * *p* < 0.05; ** *p* < 0.01; *** *p*< 0.001, as calculated by Student’s *t*-test, using data from non-obese, non-diabetic (ND) individuals as reference values. (**C**) Correlation between cAMP and FPG. (**D**) Correlation between cAMP and BMI. Correlation coefficients (r) were calculated using Pearson’s correlation test. cAMP, cyclic adenosine monophosphate; FPG, fasting plasma glucose; ND, non-diabetic; T2D, type 2 diabetes.

**Table 1 biology-09-00244-t001:** Demographic and baseline characteristics of the study population (*n* = 188).

Phenotype	T2D (*n*)	ND (*n*)	*p*-value
Age, years	52.13 ± 9.15 (92)	43.3 ± 12.782 (96)	**<0.001**
Gender (F/M)	43/49	69/27	0.143
Height, m	1.65 ± 0.08 (91)	1.63 ± 0.10 (96)	0.073
Weight, Kg	87.06 ± 14.82 (91)	78.36 ± 18.78 (96)	**0.001**
BMI, Kg/m^2^	31.60 ± 4.32 (92)	29.26 ± 5.36 (96)	**0.001**
WHR	0.96 ± 0.19 (64)	0.85 ± 0.09 (67)	**<0.001**
FPG, mmol/L	8.35 ± 3.03 (91)	5.30 ± 0.82 (94)	**<0.001**
HbA1c, %	7.75 ± 1.84 (90)	5.58 ± 0.50 (92)	**<0.001**
Total Cholesterol, mmol/L	4.96 ± 1.35 (90)	5.20 ± 0.94 (95)	0.158
HDL, mmol/L	1.19 ± 0.48 (89)	1.41 ± 0.41 (95)	**0.001**
LDL, mmol/L	3.08 ± 1.15 (88)	3.27 ± 0.87 (95)	0.226
TGL, mmol/L	1.63 ± 1.20 (90)	1.13 ± 0.67 (94)	**0.001**
Insulin, U/L	14.86 ± 10.02 (70)	9.70 ± 5.44 (57)	**<0.001**

Unless otherwise specified, data are presented as mean ± SD. *p*-Values were calculated using Student’s T-test and Fisher’s Exact test BMI, body mass index; F/M, female/male; FPG, fasting plasma glucose; HbA1c, glycated hemoglobin; HDL, high-density lipoprotein; LDL, low-density lipoprotein; *n*, number of subjects with available measurements; ND, non-diabetic; T2D, Type 2 diabetes; TGL, triglycerides; WHR, waist-to-hip ratio. Statistically significant *p*-values are in bold font.

**Table 2 biology-09-00244-t002:** Sub-analysis of demographic and baseline characteristics based on obesity status.

Phenotype	T2D			ND		
	Non-Obese (*n*)	Obese (*n*)	*p*-value	Non-obese (*n*)	Obese (*n*)	*p*-value
Age, years	51.85 ± 9.37 (34)	52.29 ± 9.10 (58)	0.827	42.00 ± 12.30 (56)	45.13 ± 13.37 (40)	0.247
Gender (F/M)	14/20	29/29-	0.517	42/14	27/13	0.492
Height, m	1.65 ± 0.07 (33)	1.66 ± 0.09 (58)	0.821	1.62 ± 0.09 (56)	1.65 ± 0.11 (40)	0.155
Weight, Kg	73.69 ± 9.00 (33)	94.66 ± 11.79 (58)	**<0.001**	67.12 ± 10.95 (56)	94.10 ± 15.96 (40)	**<0.001**
BMI, Kg/m^2^	26.80 ± 2.33 (34)	34.41 ± 2.22 (58)	**<0.001**	25.57 ± 2.89 (56)	34.43 ± 3.33 (40)	**<0.001**
WHR	0.91 ± 0.06 (22)	0.98 ± 0.23 (42)	0.077	0.83 ± 0.10 (39)	0.88 ± 0.07 (28)	**0.020**
FPG, mmol/L	7.44 ± 2.35 (33)	8.87 ± 3.26 (58)	**0.018**	5.10 ± 0.43 (54)	5.57 ± 1.11 (40)	**0.016**
HbA1c, %	6.76 ± 1.26 (32)	8.30 ± 1.88 (58)	**<0.001**	5.54 ± 0.42 (52)	5.63 ± 0.59 (40)	0.447
Total Cholesterol, mmol/L	4.94 ± 1.65 (33)	4.97 ± 1.15 (57)	0.919	5.23 ± 0.93 (55)	5.17 ± 0.95 (40)	0.770
HDL, mmol/L	1.22 ± 0.63 (33)	1.17 ± 0.36 (56)	0.658	1.46 ± 0.45 (55)	1.35 ± 0.36 (40)	0.178
LDL, mmol/L	3.12 ± 1.33 (33)	3.06 ± 1.04 (55)	0.836	3.25 ± 0.86 (55)	3.29 ± 0.89 (40)	0.799
TGL, mmol/L	1.56 ± 1.11 (33)	1.67 ± 1.26 (57)	0.652	1.06 ± 0.72 (54)	1.22 ± 0.61 (40)	0.252
Insulin, U/L	15.68 ± 11.05 (27)	14.34 ± 9.41 (43)	0.603	9.39 ± 5.94 (39)	10.38 ± 4.23 (18)	0.474

Unless otherwise specified, data are presented as mean ± SD. *p*-Values were calculated using an unequal variance *t*-test and Fisher’s exact test. BMI, body mass index; F/M, female/male; FPG, fasting plasma glucose; HbA1c, glycated hemoglobin; HDL, high-density lipoprotein; LDL, low-density lipoprotein; *n*, number of subjects with available measurements; ND, non-diabetic; T2D, Type 2 diabetes; TGL, triglycerides; WHR, waist-to-hip ratio. Statistically significant *p*-values are in bold font.

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
