# Peer review of "Increased Plasma Levels of Adenylate Cyclase 8 and cAMP Are Associated with Obesity and Type 2 Diabetes: Results from a Cross-Sectional Study"

_biology, 2020, doi:10.3390/biology9090244_

Round 1
Reviewer 1 Report
In the present study Abdel-Halim et al. showed an increase in adenylate cyclase 8 (ADCY8) protein levels in plasma from type 2 diabetes patients. cAMP levels were also increased in those patients. The authors also described a correlation between ADCY8, cAMP and classical indicators of diabetes such as HbA1c and fasting plasma glucose, respectively. Although of interest due to its novelty and its clinical relevance, there are several aspects that need to be addressed before this manuscript can be considered for publication in Biology.
General comments:
- The authors mention in the introduction that ADCYs are membrane-bound enzymes (line 48). They add then in the discussion (lines 204-206) that ADCYs might be transferred to the circulation via exosomes. Is there any literature describing the presence of ADCYs in exosomes? If so, it should be added there in the discussion.
- The female-to-male ratio is described as “significantly higher” within the T2D cohort. However, there is no statistical test proving it. I would recommend to use Fischer’s exact test and to include the p-value obtained in table 1. Likewise, the results in supplementary table 2 should be also adjusted for gender and further discuss from this perspective.
- Was the BMI self-reported? Please state this information within the material and methods.
- Please mark the P-values <0.05 within the tables in bold. That would help the reader to identify which parameters are significantly different.
- The x-axis of figures 1 and 2 only include the units. Please label them appropriate (e.g. ADCY protein levels (ng/mL)).
- The legends of figure 1 and 2 include a description of the results observed (lines 118-119 and lines 124-125), whereas the legends of figure 3 and 4 do not include this information. It is up to the authors to choose the criteria they want to follow but please make it consistent.
- Regarding the data summarized in supplementary table 3, the authors mention within the text some of the significant correlations observed (lines 141-143) but do not discuss others (for example those from ADCY3 with FPG and BMI). Was there any rationale to focus only on some specific correlations?
- When the authors analyze the combined effect of obesity and T2D on ADCY (Fig. 2A-C) and cAMP levels (Fig. 4B) a two-way ANOVA might be a more appropriate statistical test to use. This will allow to clarify whether obesity, T2D and/or the combination of both can explain the differences in ADCY and cAMP described.
Specific comments:
- Lines 33-34 and 36-37 within the abstract should be rewritten since they are hard to understand
- Likewise, lines 59-61 in the introduction can be reformulated to improve the clarity
- Line 119: a star (*) should be added before the P-value
- Line 147: please replace “instigate” by a more appropriate word
- Line 173: “levels differential protein levels” is probably a typo. Please correct it
- Line 184: please remove the comma after “variants”
- Line 198: remove the extra space before “an important”
- Line 217: remove the extra “in” and change “T2d” for “T2D”
- Line 241: remove the sign “≤”
Author Response
In the present study Abdel-Halim et al. showed an increase in adenylate cyclase 8 (ADCY8) protein levels in plasma from type 2 diabetes patients. cAMP levels were also increased in those patients. The authors also described a correlation between ADCY8, cAMP and classical indicators of diabetes such as HbA1c and fasting plasma glucose, respectively. Although of interest due to its novelty and its clinical relevance, there are several aspects that need to be addressed before this manuscript can be considered for publication in Biology.
We would like to thank the reviewer for his kind constructive feedback.
General comments:
1. The authors mention in the introduction that ADCYs are membrane-bound enzymes (line 48). They add then in the discussion (lines 204-206) that ADCYs might be transferred to the circulation via exosomes. Is there any literature describing the presence of ADCYs in exosomes? If so, it should be added there in the discussion.
We would like to thank the reviewer for his notice. We did not find a reference article describing the presence or the importance of ADCYs in plasma of exosome system. However, two proteomic studies listed ADCY10, ADCY1 and ADCY4 in the supplementary tables with no further details in the main article body. We referenced the two articles in the discussion section in support to our hypothesis (Lines: 230-236, highlighted in green).
A. ADCY10 was listed in supplementary data file:
In a protomeric profiling of exosomes purified from human plasma, Smolarz and his colleagues purified plasma exosomes, from four healthy individuals, using size-exclusion chromatography (SEC) followed by untargeted LC–MS/MS proteomic analysis. The authors enriched exosomes fractions using CD36 and CD81 markers. ADCY10 peptides were detected in the exosome’s fractions 7 and 9, however, the author did not mention this in the article.
https://doi.org/10.3390/proteomes7020018
B. ADCY1 and 4 were listed at the supplementary data file:
Wang and his team studied the proteomic profile for extracellular vesicles isolated from 50 urine specimens that were biobanked at the Parkinson’s Disease Biomarkers. The authors were interested to identify markers for Parkinson Disease. In supplementary Table 3, ADCY1 and ADCY4 were differentially regulated PD samples relative to controls, but the authors did not mention this observation in their article.
https://doi.org/10.1016/j.ebiom.2019.06.021
2. The female-to-male ratio is described as “significantly higher” within the T2D cohort. However, there is no statistical test proving it. I would recommend to use Fischer’s exact test and to include the p-value obtained in table 1. Likewise, the results in supplementary table 2 should be also adjusted for gender and further discuss from this perspective.
We would like to thank the reviewer for his advice.
A. As advised by the reviewer Fisher’s Exact test was performed and no significant difference in the distribution of gender was observed across the tested subjects. Hence, the statement is corrected as follows: “The female-to-male ratio was higher among individuals without T2D than among those with T2D” (lines: 83-84, highlighted in green).
B. Fisher’s Exact test and P-values were included in Table 1
C. Supplementary Table 2 has been adjusted for both age and gender, as advised. The outcome of the adjustments to age and gender were listed at paragraph 2.3, Lines 140-150.
3. Was the BMI self-reported? Please state this information within the material and methods.
We would like to thank the reviewer for his advice.
BMI was determined by a specialized nurse at DDI facility using the standard formula for the body mass index, BMI = body weight (kg) / height2 (m2). The statement has been corrected in the Material and Methods section of the manuscript (Lines: 265-267, highlighted in green).
4. Please mark the P-values <0.05 within the tables in bold. That would help the reader to identify which parameters are significantly different.
As advised by the reviewer, P-values <0.05 within the Tables has been bolded.
5. The x-axis of figures 1 and 2 only include the units. Please label them appropriate (e.g. ADCY protein levels (ng/mL)).
As advised by the reviewer, the Y-axis of Figures 1 and 2 have been edited and labelled appropriately.
6. The legends of figure 1 and 2 include a description of the results observed (lines 118-119 and lines 124-125), whereas the legends of figure 3 and 4 do not include this information. It is up to the authors to choose the criteria they want to follow but please make it consistent.
As notified by the reviewer, the Figure legends have been modified appropriately.
7. Regarding the data summarized in supplementary table 3, the authors mention within the text some of the significant correlations observed (lines 141-143) but do not discuss others (for example those from ADCY3 with FPG and BMI). Was there any rationale to focus only on some specific correlations?
We would like to thank the reviewer for his notice.
We wanted to focus on ADCY8, but the reviewer is right. When the whole population was studied, we did not detect correlations between ADCY3 and the tested biomedical parameters, however, when the data are adjusted to age and gender as suggested by the reviewer, we observed an inversed correlation between ADCY3 and BMI (Suppl Table 2). When data was stratified with obesity, the inverse correlation was observed in obese individuals. These data are in accordance with two population studies identifying loss-of-function mutations within ADCY3 gene in obese and T2D patients (PMID: 29311636 and PMID: 29311637, both are cited in the manuscript). We have added this valuable notice from the reviewer to the result section (Lines 147-150, highlighted in green) and the discussion section (lines 203-206, highlighted in green). Supplementary Table 2 with adjustments to age and gender.
8. When the authors analyze the combined effect of obesity and T2D on ADCY (Fig. 2A-C) and cAMP levels (Fig. 4B) a two-way ANOVA might be a more appropriate statistical test to use. This will allow to clarify whether obesity, T2D and/or the combination of both can explain the differences in ADCY and cAMP described.
We would like to thank the reviewer for his notice.
As advised by the reviewer, a Two-way ANOVA with the post-hoc Tukey test was conducted to further examine the effect of diabetes and obesity on plasma ADCYs. ADCY8 was observed to be significantly higher in T2D compared to ND (P = 0.04). More profoundly diabetic obese individuals showed significantly higher expression of ADCY8 compared to the control group (P = 0.02). No such observations were made in case of ADCY1 and ADCY3 (Section 2.2 was modified, ANOVA data was incorporated,Llines: 90-106).
Likewise, obesity status of the participant tends to more profoundly influence the levels of cAMP (P > 0.008); (Section 2.4, Lines: 171-173).
Reviewer 2 Report
To the Authors:
In this study the authors investigate the levels of plasma Adenylate Cyclase 8 and cAMP from obese, non-obese, type 2 diabetic and non-diabetic human cohorts. Although the authors show that ADCY8 is increased in obese T2D compared with obese non-diabetics and lean T2Ds individuals, the interpretation of this data is currently premature in its current form. The authors also suggest that ADCYs may be a valuable marker for the prediction of T2Ds. The study is not designed to answer this question, and based on these results, will not be as efficient biomarker as FPG or %HbA1c. These points are not discussed and I believe the manuscript needs to be rewritten in a way to reflect the findings. That is, to highlight the phenotype that leads to increased ADCYs, why this is, and what is the potential outcomes or effects of this increased ADCYs. To report this as a potential biomarker and to further suggest it may help predict T2D is incorrect based on the current results. In short, the manuscript needs to be written in a way that reflects the findings from the given data.
- Firstly the authors claim that that ADCY play a role in T2D and those at risk of developing T2D (line 66), the study is not designed to answer this question. The only cohort that are at risk of developing T2D are the ND obese, who display no changes in ADCY’s in the data provided, suggesting that ADCY,s are not pre-emptive to diabetes, but rather a result of T2D. Further to this, the data provided actually suggest that these changes may actually be linked to obesity rather than T2Ds on its own.
- Further breakdown of the data may help tease out some of the results. In particular, additional figures/tables that break down the data into males and females, such as figures 1, 2, & 3. The authors already have this data, so it would be beneficial to investigate this further.
- Some discussion and interpretation on fig2B: why are both the T2D data less variable? What does this mean?
- Fig4B suggests that the increase in cAMP may have a stronger correlation with obesity, not T2Ds, as mentioned above in regards to ADCYs. Is it a result of hyperinsulinemia?
- Line 217-218: Again, this is not a true biomarker of T2Ds, based on these results. To make these claims a study that takes measurement of ADCYs through disease progression is required.
Author Response
In this study the authors investigate the levels of plasma Adenylate Cyclase 8 and cAMP from obese, non-obese, type 2 diabetic and non-diabetic human cohorts. Although the authors show that ADCY8 is increased in obese T2D compared with obese non-diabetics and lean T2Ds individuals, the interpretation of this data is currently premature in its current form. The authors also suggest that ADCYs may be a valuable marker for the prediction of T2Ds. The study is not designed to answer this question, and based on these results, will not be as efficient biomarker as FPG or %HbA1c. These points are not discussed and I believe the manuscript needs to be rewritten in a way to reflect the findings. That is, to highlight the phenotype that leads to increased ADCYs, why this is, and what is the potential outcomes or effects of this increased ADCYs. To report this as a potential biomarker and to further suggest it may help predict T2D is incorrect based on the current results. In short, the manuscript needs to be written in a way that reflects the findings from the given data.
We would like to thank the reviewer.
A. Yes, we agree with the reviewer and appreciate his valuable input. We do not have data to support the fact that ADCYs are biomarkers for T2D, accordingly, we have deleted the statements of biomarker.
B. Accordingly, we have a adjusted the manuscript to reflect the role of obesity on the plasma level of the studied ADCYs.
C. The experimental design was to investigate the potential availability of plasma ADCYs; identify associations between plasma ADCYs, obesity and diabetes; and to correlate the protein levels of plasma ADCYs with different clinical parameters.
D. A follow up study through disease progression is required to claim its true potential, currently, we are recalling the participants for longitudinal follow up studies.
1. Firstly the authors claim that that ADCY play a role in T2D and those at risk of developing T2D (line 66), the study is not designed to answer this question. The only cohort that are at risk of developing T2D are the ND obese, who display no changes in ADCY’s in the data provided, suggesting that ADCYs are not pre-emptive to diabetes, but rather a result of T2D. Further to this, the data provided actually suggest that these changes may actually be linked to obesity rather than T2Ds on its own.
We would like to thank the reviewer for this important comment.
A. The presented study was based on our original observation in a series of studies of overexpression of ADCY3 and 8 in an experimental animal model of spontaneous T2D and whether this can be reflected in the plasma as a marker for the disease. In human tissues, ADCY8 has also been reported to be associated with either T2D or obesity (Sung, Y.J, et al., doi: 1038/ijo.2015.217 ). However, no one has established a study to quantify ADCYs levels in human plasma. We agree with the reviewers that ADCY8 is associated with obesity and T2D factors.
B. Currently, we are following the patients for a long period of time in longitudinal study to see the relevance of ADCYs with the diabetes, obesity and their complications. Hopefully, we will report in a couple of years as a follow up. Our results may help understanding the clinical implications of plasma ADCYs. levels in the future.
C. We have removed the statements that ADCYs may act as biomarkers from the manuscript and added that further studies are currently underway in our laboratory to understand the consequences of high ADCY 8 plasma levels in T2D in terms of cardiac arrythmias and other complications.
2. Further breakdown of the data may help tease out some of the results. In particular, additional figures/tables that break down the data into males and females, such as figures 1, 2, & 3. The authors already have this data, so it would be beneficial to investigate this further.
We thank reviewer for this remarkable insight.
The influence of gender on ADCYs in whole population has been shown in Supplementary Table 1. We did not see significant differences in the level of ADCYs with respect to gender. In addition, when the data was adjusted to age and gender, minimal effect was observed (Supplementary Table 2). We have added this valuable notice from the reviewer to the result section (Lines 147-150, highlighted in green) and the discussion section (Lines 203-206, highlighted in green). Supplementary Table 2 with adjustments to age and gender.
3. Some discussion and interpretation on fig2B: why are both the T2D data less variable? What does this mean?
We would like to thank the reviewer for the advice
As shown in Table 1, there is a significant difference in the distribute of age across T2D and ND (P <0.001). However, a significantly positive correlation was observed between ADCY3 and age in ND individual, but not in T2D patients, Supplementary Table 3. Therefore, it is possible that the age factor may justify the less variability observed in T2D patients (Lines 108-112, highlighted in green).
4. Fig4B suggests that the increase in cAMP may have a stronger correlation with obesity, not T2Ds, as mentioned above in regard to ADCYs. Is it a result of hyperinsulinemia?
We would like to thank the reviewer for the advice
Fig4a shows clear a significant increase of cAMP in T2D patients. That increase was even more pronounce obese-T2D. The reviewer raised a very important point which show that in ND obesity induces a higher level of cAMP. Indeed, this may due to hyperinsulinemia (Lines 171-173, Highlighted in green).
5. Line 217-218: Again, this is not a true biomarker of T2Ds, based on these results. To make these claims a study that takes measurement of ADCYs through disease progression is required.
We would like to thank the reviewer for the advice
A. We have removed the statements that ADCYs may act as biomarkers from the manuscript and added that further studies are currently underway in our laboratory to understand the consequences of high plasma levels of ADCY 8 in obese-T2D in terms of cardiac arrhythmias and other complications.
B. The measures of high plasma ADCY8 and its correlation with BMI and HbA1c might be crucial in a clinical setting, however we agree with the reviewer a follow up study through disease progression is required to claim its true potential. Currently, we are recalling the participants for longitudinal follow up studies. Conclusion has been modified accordingly (discussion section, Lines 214-229; conclusion section Lines 250-256).
Round 2
Reviewer 2 Report
The authors have adequately addressed my concerns
Author Response
We would like to thank the reviewer for his kind impact which has tremendously improved our manuscript.
Before the submission of the revised version, we sent the manuscript for English proof and editing though an international corporate. Currently, the one spilling error notice is
Line 87: compated corrected to compared
If there are others, hopefully, during the proof-reading process will be corrected.
The comment about “research design appropriate, can be improved”: We are happy to answer the reviewer and improve the manuscript. Can you specify exactly what the reviewer is asking about? What are the specific points that the reviewer wants us to change or expand?